# Development and Validation of an Explainable Radiomics Model to Predict High-Aggressive Prostate Cancer: A Multicenter Radiomics Study Based on Biparametric MRI

**DOI:** 10.3390/cancers16010203

**Published:** 2024-01-01

**Authors:** Giulia Nicoletti, Simone Mazzetti, Giovanni Maimone, Valentina Cignini, Renato Cuocolo, Riccardo Faletti, Marco Gatti, Massimo Imbriaco, Nicola Longo, Andrea Ponsiglione, Filippo Russo, Alessandro Serafini, Arnaldo Stanzione, Daniele Regge, Valentina Giannini

**Affiliations:** 1Department of Electronics and Telecommunications, Polytechnic of Turin, Corso Duca degli Abruzzi, 24, 10129 Turin, Italy; giulia.nicoletti@polito.it; 2Department of Surgical Sciences, University of Turin, Corso Dogliotti, 14, 10126 Turin, Italy; valentina.cignini@unito.it (V.C.); riccardo.faletti@unito.it (R.F.); alessandro.serafini@unito.it (A.S.); 3Radiology Unit, Candiolo Cancer Institute, FPO-IRCCS, Strada Provinciale, 142—KM 3.95, 10060 Candiolo, Italy; simone.mazzetti@ircc.it (S.M.); giovanni.maimone@unito.it (G.M.); filippo.russo@ircc.it (F.R.); daniele.regge@ircc.it (D.R.); 4Department of Medicine, Surgery, and Dentistry, University of Salerno, Via Salvador Allende, 43, 84081 Baronissi, Italy; rcuocolo@unisa.it; 5Department of Advanced Biomedical Sciences, University of Naples “Federico II”, Via Pansini, 5, 80131 Naples, Italy; massimo.imbriaco@unina.it (M.I.); a.ponsiglionemd@gmail.com (A.P.);; 6Department of Neurosciences, Reproductive Sciences and Odontostomatology, University of Naples “Federico II”, Via Pansini, 5, 80131 Naples, Italy; nicola.longo@unina.it; 7Department of Translational Research, Via Risorgimento, 36, University of Pisa, 56126 Pisa, Italy

**Keywords:** radiomics, prostate cancer, magnetic resonance imaging, feature extraction, explainable artificial intelligence, tumor aggressiveness

## Abstract

**Simple Summary:**

Prostate cancer (PCa) is one of the leading causes of mortality for men worldwide. PCa aggressiveness affects the patient’s prognosis, with less aggressive tumors, i.e., Grade Group (GG) 1 and 2, having lower mortality and better outcomes. For this reason, the aim of this study is to distinguish between GG ≤ 2 and ≥3 PCa using an automatic and noninvasive approach based on artificial intelligence methods. The results obtained are promising, as the system achieved robust results on a multicenter external dataset. If further validated, this approach, combined with the expert knowledge of urologists, could help identify PCa patients who have a better prognosis and may benefit from less invasive treatments.

**Abstract:**

In the last years, several studies demonstrated that low-aggressive (Grade Group (GG) ≤ 2) and high-aggressive (GG ≥ 3) prostate cancers (PCas) have different prognoses and mortality. Therefore, the aim of this study was to develop and externally validate a radiomic model to noninvasively classify low-aggressive and high-aggressive PCas based on biparametric magnetic resonance imaging (bpMRI). To this end, 283 patients were retrospectively enrolled from four centers. Features were extracted from apparent diffusion coefficient (ADC) maps and T2-weighted (T2w) sequences. A cross-validation (CV) strategy was adopted to assess the robustness of several classifiers using two out of the four centers. Then, the best classifier was externally validated using the other two centers. An explanation for the final radiomics signature was provided through Shapley additive explanation (SHAP) values and partial dependence plots (PDP). The best combination was a naïve Bayes classifier trained with ten features that reached promising results, i.e., an area under the receiver operating characteristic (ROC) curve (AUC) of 0.75 and 0.73 in the construction and external validation set, respectively. The findings of our work suggest that our radiomics model could help distinguish between low- and high-aggressive PCa. This noninvasive approach, if further validated and integrated into a clinical decision support system able to automatically detect PCa, could help clinicians managing men with suspicion of PCa.

## 1. Introduction

Prostate cancer (PCa) is the most common cancer in men in Western countries, accounting for about 25% of new cancer diagnoses [1,2]. Since 2020, the European Association of Urology added a strong recommendation to perform multiparametric magnetic resonance imaging (mpMRI) in PCa patients before either planning a re-biopsy or in biopsy-naïve men [3] to better identify PCa suspicion and to target the biopsy to retrieve more precise information about cancer aggressiveness [4]. However, biopsies are known to suffer from limitations that negatively impact the therapeutic path and patients’ outcomes [5]. Indeed, they are known to be invasive, affected by large inter-observer variability, and not able to provide information on spatial tumor heterogeneity [6,7].

In the last years, radiomics analysis to characterize PCa has been demonstrated as a potential alternative to overcome the main biopsy drawbacks [8,9]. The radiomics approach is challenging, and researchers have mainly focused on the identification of clinically significant ones (GG ≥ 2) [10], reaching considerably high performances [11]. However, a more precise separation between GG2 and GG3 is necessary since GG3 PCas show more aggressive behavior, with higher rates of biochemical failure, systemic recurrence, cancer-specific death [12,13,14], and lower probability of 5-year biochemical risk-free survival [4]. Only a few studies have classified GG ≤ 2 and GG ≥ 3 PCas [15,16,17,18] through radiomics analyses, but the generalization of their results has not been fully addressed yet. Moreover, these studies did not adopt an explainable approach [15,16,17,18]. A not-explained approach limits transparency, trustworthiness, and, therefore, application in real-world clinical practice [19]. To our knowledge, there is no study validating a machine learning classifier to predict GG ≤ 2 and GG ≥ 3 PCas on an external validation set and proposing an explainable AI approach. 

The aims of this study are to develop a model to noninvasively distinguish between GG ≤ 2 and GG ≥ 3 PCa through a radiomics signature based on bpMRI and to validate this model on a multicenter dataset composed of images acquired with different MRI manufacturers, including both 1.5T and 3T scanners, also providing an explanation for the signature. The proposed pipeline is noninvasive, as it uses already available acquisitions from MRI without endorectal coil and contrast agent administration. Our model showed robust results on the external validation set, suggesting that, if further validated, it could be used as a noninvasive tool to select low-aggressive PCas, which, if confirmed by additional clinical evaluations, could avoid destructive and invasive treatments. 

## 2. Materials and Methods

### 2.1. Patients

This is a multicenter retrospective study that includes four institutes: University Hospital “Città della Salute e della Scienza” of Turin (center A), “Mauriziano Umberto I” Hospital of Turin (center B), Candiolo Cancer Institute (center C), and Federico II Hospital of Naples (center D). Patients who underwent prostate MRI between November 2015 and October 2018 at sites A, B, and D and between May 2017 and February 2020 at site C were included in the study. The following inclusion criteria were applied: (1) patients with suspicion of PCa; (2) patients who underwent MRI examination before biopsy, including at least diffusion-weighted (DWI) and T2-weighted (T2w) axial images; and (3) imaging performed without endorectal coil. Exclusion criteria were as follows: (1) the presence of strong artifacts on the MRI examination; (2) patients who underwent transurethral resection of the prostate (TURP); (3) no match between the tumor location detected by the biopsy and the MRI’s findings; and (4) absence of pathologically confirmed (either biopsy or prostatectomy) PCa.

### 2.2. MRI Acquisition and Reference Standard

MRI examinations were performed using 1.5T scanners at centers A, B, and C (Achieva Philips Medical Systems, Ingenia Philips Medical Systems, and Optimae GE Healthcare, respectively) and a 3T scanner at center D (Magnetom Trio Siemens Medical Solutions). The reference standard for this study was the GG obtained after targeted biopsy or prostatectomy, when available, on the lesions detected on MRI. More information on scanner parameters can be found in Appendix A. For patients with available prostatectomy results, the exact location of the tumor was found by comparing the MRI with the microscopic slices of the surgical specimen [19]. When the results of the prostatectomy were not available, the tumor was localized on MRI using the detailed report provided by uropathologists [20]. Dedicated uropathologists examined the hematoxylin- and eosin-stained slides and recorded the GG. Finally, PCas were dichotomized into low-(GG ≤ 2) and high-(GG ≥ 3) aggressive cancers, hereafter considered as the negative and positive class, respectively.

### 2.3. Tumor Segmentation and Feature Extraction

Four radiologists (one for each of the recruiting centers) with more than 5 years of experience manually segmented on T2w imaging all PCas using ITK Snap 3.8 [21] (www.itksnap.org, accessed on 29 December 2023), and the available contouring or report provided by uropathologists as a reference, as previously described in [22]. During the segmentation step, the radiologist checked that the mask of the tumor also matched the tumor on the ADC maps, and in case of misalignment, they reviewed the mask to segment only the common areas, as previously reported [22]. Then, an experienced senior radiologist reviewed and eventually corrected all the segmentations from the four centers. After a step of image preprocessing where both T2w sequences and ADC maps were normalized and pixels showing outlier signal intensities were removed, 169 first and second-order features were extracted (more details are available in Appendix A). Features were extracted in Python 3.8 using the open-source Python package Pyradiomics 3.0.1 [23], compliant with the Image Biomarker Standardization Initiative [24]. Figure 1A reports the main steps from image acquisition to feature extraction.

### 2.4. Model Development and Validation

During the development of the radiomics signature, 26 combinations of feature selection (FS) methods and classifiers/regressors were fine-tuned using only patients from centers A and B (construction dataset). Specifically, the feature selection methods included (1) minimum redundance maximum relevance, (2) affinity propagation, feature ranking based on (3) chi-squared test or (4) Mann–Whitney U test, and (5) stepwise binomial logistic regressor. These 5 FS methods were combined with the following classifiers/regressors: (1) decision tree, (2) support vector machine, (3) ensemble learner (e.g., random forest), (4) naïve Bayes, and (5) binomial logistic regression. In addition, we evaluated the LASSO logistic regressor. Before performing the FS step, all features were normalized between 0 and 1 using the min-max scaling. The radiomics pipeline is reported in Figure 1. All algorithms were implemented in MATLAB^®^ 2021b. 

To select the best-performing model, a cross-validation (CV) strategy has been adopted, in which 4 folds (training a stratified 5-fold cross-set) were iteratively used to train a model that was subsequently tested on the left-out fold (test set). During the CV, all the combinations of FS and classifiers were evaluated. The parameters of the models were tuned, and models not reaching a mean area under the receiver operating characteristic (ROC) curve (AUC) ≥ 0.6 on the left-out folds were discarded. More details about the FS methods and classifiers are described in Appendix A. 

Once all the FS/classifier/parameters combinations were tested, the one reaching the highest mean AUC on the left-out folds was selected and re-trained using the whole construction set (Figure 1B).

The validation step was performed by applying the previously selected model on the validation set, which included only patients that were left out from the model development phase, i.e., from centers C and D (Figure 1C). We decided to use centers A and B as the construction set to develop the model on a strong reference standard since all their patients had the prostatectomy results available and to train a model on 1.5T MRI images and evaluate its generalization capability on both 1.5T and 3T (center D) MRI images. The validation set was normalized using the min-max scaling, with the maximum and minimum values of the construction set.

### 2.5. Statistical Analysis

Correlation between all pairs of features employed in the best model was computed using Spearman’s correlation test (MATLAB^®^ 2022b). Balanced accuracy, sensitivity, specificity, positive predictive value (PPV), and negative predictive value (NPV) obtained using the cut-off corresponding to the Youden Index on the construction set were computed as an example of general model performances. Sensitivity was defined as the number of correctly classified high-aggressive PCas over the total number of high-aggressive PCas; specificity was defined as the number of correctly classified low-aggressive PCas over the total number of low-aggressive PCas; NPV was defined as the number of correctly classified low-aggressive PCas over the total number of patients classified as low-aggressive PCas; and PPV was defined as the number of correctly classified high-aggressive PCas over the total number of patients classified as high-aggressive PCas.

For the 5-fold CV, the difference in the mean of the six performance indexes between train and test sets was calculated to evaluate the robustness of the models and exclude those combinations that tend to overfit. Since in the literature, there is no commonly accepted criterion to identify overfitting models, in this analysis, a decrease in performances from the training set to the test set greater than 30% was considered overfitting. For the final classifier, changes in the performances obtained between the construction set and the validation set and between centers C and D were evaluated using the N−1 chi-squared test performed for each of the performance metrics, while AUCs were compared based on the DeLong test. A *p*-value < 0.05 was considered statistically significant. All the statistical analyses were computed on MedCalc Statistical Software version 20.105 (MedCalc Software bv, Ostend, Belgium).

### 2.6. Explanation of the Classification Model

To better understand the signature of the final radiomics model and the role of the selected features in the classification task, Shapley additive explanation (SHAP) values were computed for the external validation set. The SHAP values of a feature explain the role of that feature in “pushing” the model output toward positive or negative predictions [25]. Then, the feature importance was calculated as the average of the absolute SHAP values per feature across the external validation set. In addition to features explanation, we provided a global interpretation of the signature by computing the partial dependence plot (PDP) of each feature employed in the model. The PDP displays the marginal effect that a feature has on the predicted outcome, showing whether the relationship between the output and that feature is linear or more complex (see also Appendix A).

## 3. Results

### 3.1. Patient Characteristics

The multicenter dataset included a total of 299 PCas from 283 patients. The construction set included 175 PCas (132 from A and 43 from B, 77 low- and 98 high-aggressive), while the validation set was composed of 124 PCas (78 from C and 46 from D, 69 low- and 55 high-aggressive). More details are described in Figure 2. GG was derived from prostatectomy in 86% (243/283) of patients, while in the remaining patients, we used the GG provided by targeted biopsy. The review of the segmentations performed by the expert senior radiologist resulted in less than 10% of changes in the tumor masks provided by the four centers.

### 3.2. Best Model

The naïve Bayes classifier trained using ten features selected by the affinity propagation algorithm was chosen as the best model based on its AUC of the left-out folder (Section 2.4) (see Appendix A for more details). The Spearman correlation coefficient (r) for the different combinations of the ten selected features showed a fair correlation (r ≤ 0.50) [26] for 40/45 pairs and a moderate correlation (0.50 < r ≤ 0.70) only in 5/45 pair comparisons. No highly correlated features, i.e., higher than 0.70, were found in the selected subset. The results of the final classifier trained with all patients enrolled from centers A and B and then externally validated with the cases from centers C and D are reported in Table 1. On the validation set, the classifier achieved results comparable to those obtained on the construction set, i.e., AUC of 0.75 and 0.73, respectively. No significant differences were found in the performance metrics between the construction and external validation sets (Table 1). In the comparison of the performances between centers C and D, only specificity was significantly lower in center D (*p*-value < 0.05) (Table 1). Figure 3 shows the discrete non-smoothed ROC curves of the construction and validation set. For completeness and transparency of reports of the diagnostic accuracy of this study, we reported the Standards for Reporting of Diagnostic Accuracy Studies (STARD) diagram [27] in Appendix A. The waterfall plots of the output signature of the classifier for the construction and validation sets are displayed in Figure 3. As we can see, the classifier is highly accurate when assigning a likelihood of a high-aggressive tumor equal to zero (<1%). Indeed, 11 out of 15 lesions of the training set (Figure 4A) and all 15 lesions of the validation set (Figure 4B) with assigned likelihood equal to zero are true low-aggressive PCas.

### 3.3. Explanation of the Best Model

Figure 5 displays the bar diagram reporting features in decreasing order of importance. The ‘ADC-GLRLM- Run Length Non-Uniformity’ was the most important feature, changing the predicted high-aggressive PCa probability, on average, by 14.5 percentage points, followed by the ‘ADC-GLRLM- Run entropy’ changing the prediction, on average, by 7.2 percentage points. The SHAP summary plot is reported in Figure 6, displaying on the y-axis all features of the NB classifier ordered by importance and on the x-axis their corresponding SHAP values, with color representing the value assumed by the feature, from low (red) to high (blue). It can be seen how the highest absolute SHAP values of the first most important feature were positive, meaning that the feature contributed more to the high-aggressive PCa predictions than to the low-aggressive ones. Vice versa, the second most important feature is characterized mainly by negative SHAP values, highlighting that it mainly contributed to the low-aggressive PCa predictions. Interestingly, low values of ‘ADC-GLRLM- Run entropy’ reduce the predicted high-aggressive PCa risk. In Figure 7, a local visualization of the SHAP values of a true positive and true negative prediction randomly selected from the external validation set is shown. As a corroboration of what has been deduced from the SHAP summary plot, the local visualization shows how, in the case of a true positive prediction (Figure 7a), the most important feature in Figure 5 has the highest SHAP value (0.434), thus highlighting its relevant contribution to that final prediction of a high-aggressive PCa. Similarly, for the true negative prediction example (Figure 7b), the second most important feature contributed far more than the others to the final classification, decreasing the predicted high-aggressive PCa probability of 24 percentage points, thus pushing the predictions toward the low-aggressive class.

## 4. Discussion

In this study, we developed and validated a radiomics model to distinguish between GG ≤ 2 and GG ≥ 3 PCas using bpMRI. Regarding the classification GG ≤ 2 and GG ≥ 3, to the best of our knowledge, this is the first study that externally validated a classification model and that provided an explanation for the model’s output. External validation is a challenging task since MRI suffers from high variability due to differences in scanners and acquisition protocols and researchers are struggling to develop AI-based models dealing with multicenter datasets. However, thanks to the engineering of the pipeline that we used to obtain the most robust and generalizable model, we reached promising results on both the construction and the external validation set, which included different scanners and magnetic field strengths. An important element of our system is that we provided, together with the binary classification, the likelihood of the tumor being high-aggressive. The latter can be considered to tune the classification output to maximize either NPV or PPV. Considering the value of this score, our studies showed an important result: all 15 lesions in the validation set having a likelihood of being high-aggressive equal to zero were indeed either GG1 or GG2 lesions. If further validated on a larger dataset, this might impact the management of this subgroup of patients for whom an invasive procedure might be avoided.

In the literature, several studies developed AI-based algorithms to characterize PCa aggressiveness, mainly focusing on distinguishing clinically significant PCas (GG1 vs. GG ≥ 2) [10] and reaching promising results in multicenter datasets [11]. However, only a few studies focused on the distinction of GG1/GG2 lesions from higher aggressive PCas, obtaining a cross-validated AUC from 0.75 to 0.77 using different machine learning algorithms [16,17]. Cuocolo et al. [18] evaluated the relationship between shape features and low/high aggressive tumors through a multivariable logistic analysis, reaching an AUC of 0.78 with a specificity and sensitivity of 97% and 56%, respectively, on the training set. Bertelli et al. [15] developed an ensemble learner classifier obtaining an AUC of 0.79 on an internal validation set, i.e., using a monocenter dataset. However, all these studies included only one center in their dataset, and none of them validated their results on an external set. Our results are comparable to those in the literature but with the remarkable advantage of keeping two centers as an external validation set, demonstrating the generalization capability of the radiomics classifier (no significant differences were found in the performances of the construction and validation set). As an additional strength of this work, we obtained a radiomics quality score (RQS) [8] of 11 (Appendix A), which is higher than the average RQS score of 7.93 previously calculated for prostate radiomics studies [28].

Finally, the innovative side of this study is that we have provided an explainable model, deriving relevant information about the importance of selected features. More specifically, seven out of ten features used by the best model were extracted from the ADC map, including the ADC mean, whose value was demonstrated to have a negative relationship with the probability of high-aggressive PCa (Figure 6 and Appendix A). This is consistent with the literature since ADC values are demonstrated to be moderately correlated to GG of peripheral zone lesions [29] and to have a role in differentiating GG2 and PCa with higher grades [16,30,31,32]. Moreover, it was demonstrated that high-aggressive peripheral zone tumors are associated, on the ADC map, with high values of entropy and low values of energy [33]. Interestingly, considering features individually, we found that entropy plays a crucial role in distinguishing between low and high aggressive PCas, also in our algorithm. Indeed, the ‘ADC–GLRLM–Run Entropy’, the second feature in decreasing order of importance (Figure 5) that measures the uncertainty/randomness in the distribution of run lengths, i.e., sequences in a straight scan direction of pixels with identical image value, and gray levels, was found to play a relevant role in the prediction of low-aggressive PCas. Low values of this feature, i.e., high homogeneity, were found to push predictions towards a lower probability of high-aggressive PCas (Figure 6). Conversely, the most important feature of the model, the ‘ADC-GLRLM- Run Length Non-Uniformity’ (Figure 5), drives the classification of PCa towards the high-aggressive class.

Nevertheless, our study has some limitations. The first limitation regards the limited sample size of the four centers, which affected several aspects of the results of this study. First, the sample size of center D not only impacts the robustness of the results of center D, but it might also be responsible for the significant difference in specificities that we found between centers C and D of the validation set. Indeed, it is important to notice that center D includes only 20 low-aggressive lesions, and therefore, the obtained *p*-value, close to 0.05, is highly influenced by the sample size. An increase in the number of low-aggressive lesions acquired with a 3T scanner would be beneficial to either confirm or reject the hypothesis that specificities between 1.5T and 3T datasets are significantly different. Second, an increase in the sample size of all four centers would have allowed us to perform cross-validation, permuting two centers in the construction set and then externally validating the models on the two remaining ones. This permutation strategy was implemented as our first attempt at the radiomic pipeline, but it did not result in acceptable performances, probably due to the low sample size and the unbalanced number of patients per class across the different centers. Third, we found that all of the 15 lesions of the validation set with a null likelihood of being high-aggressive were indeed GG ≤ 2; however, this was not true in the construction set, where 4/15 lesions with a null likelihood of being high-aggressive were misclassified. This difference may be due to the number of GG1 PCa in the validation set (n = 20), which was higher than in the construction set (n = 1). Therefore, an increase in the sample size of the centers would be of key importance to confirm the generalizability of the model in classifying low-aggressive PCas. A second limitation is that we did not perform a preliminary step of detection of outliers, which affected the values of some features. As can be seen in the colors of the SHAP plot of the construction and validation sets, since the construction set contained some elements with particularly high or low values of some features, after min-max normalization, these features were skewed towards, respectively, low or high values. However, we believe that this does not affect the predictions since for those methods using the distribution of values (e.g., Bayesian), the distribution is not impacted by the presence of one/two outliers, while for those methods using hyperplanes (e.g., SVM), again, the position of the hyperplanes is not influenced by the presence of a few outliers. A third limitation is that the reference standard of 14% of patients was based on target biopsy. However, the model was trained using only patients having prostatectomy as the reference standard; therefore, bias might be introduced only in the validation set. Finally, we did not account for intra-lesion heterogeneity to stratify GG2 and GG3 PCas according to the percentage of Gleason pattern 4. In the future, it would be useful to provide a probability map of tumor grade heterogeneity together with the aggressiveness index to spatially characterize different tissue characteristics.

## 5. Conclusions

In this study, we developed a radiomics model, based on texture features from bpMRI, that automatically assigns a PCa aggressiveness class to selected suspicious lesions, distinguishing tumors with a good prognosis, i.e., low-aggressive PCas, from more aggressive ones. Predicting high-aggressive PCa with radiomics remains very challenging; however, this noninvasive approach, if further validated and integrated into a clinical decision support system, could help clinicians to manage men with suspicion of PCa, suggesting personalized treatments and selecting patients that might benefit from radical treatment and those that could enter surveillance protocols or undergo less destructive treatments, avoiding biopsy discomfort and related complications.

## Figures and Tables

**Figure 1 cancers-16-00203-f001:**
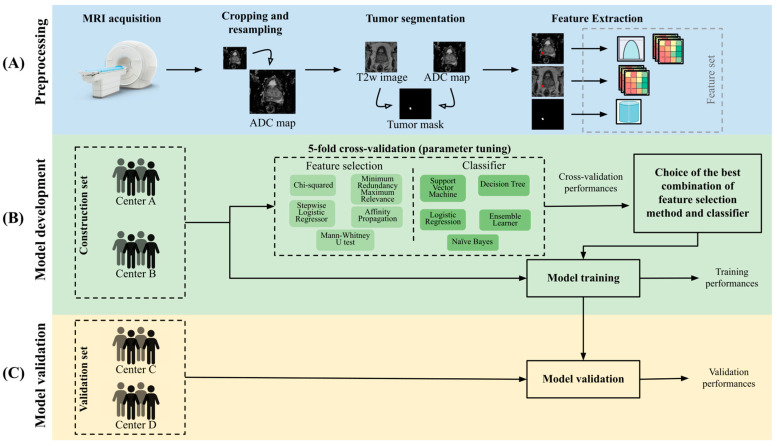
Pipeline of the study including (**A**) preprocessing, (**B**) model development, and (**C**) validation steps. ADC = apparent diffusion coefficient, T2w = T2-weighted, MRI = magnetic resonance imaging.

**Figure 2 cancers-16-00203-f002:**
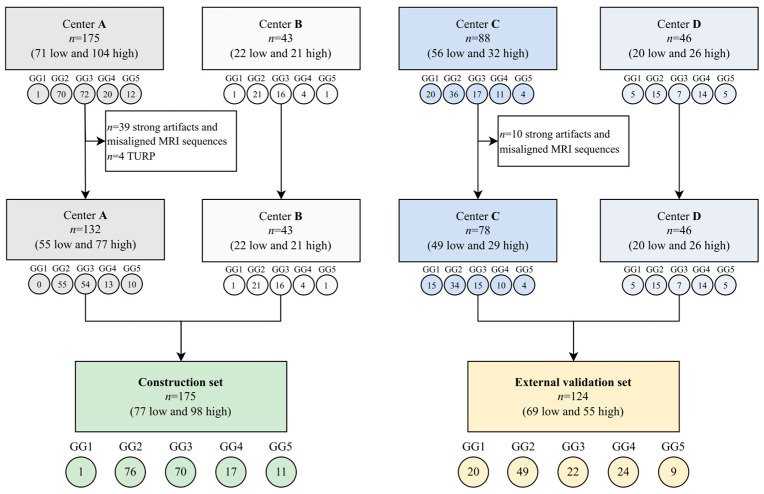
Flowchart of dataset division. In each box: N is the total number of PCas; in parentheses are reported the number of low-and high-aggressive PCas. Under each box: the number of lesions for each of the 5 GGs is reported in the circles. GG = grade group, MRI = magnetic resonance imaging, TURP = transurethral resection of the prostate.

**Figure 3 cancers-16-00203-f003:**
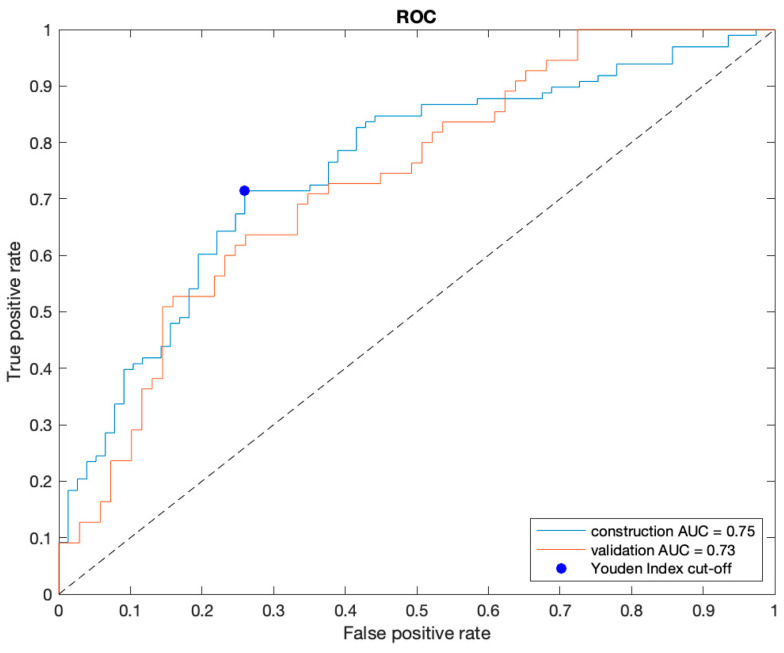
Receiver operating characteristic (ROC) curve of construction (blue) and validation set (red). The blue point corresponds to the cut-off based on the Youden Index. AUC = area under the ROC curve.

**Figure 4 cancers-16-00203-f004:**
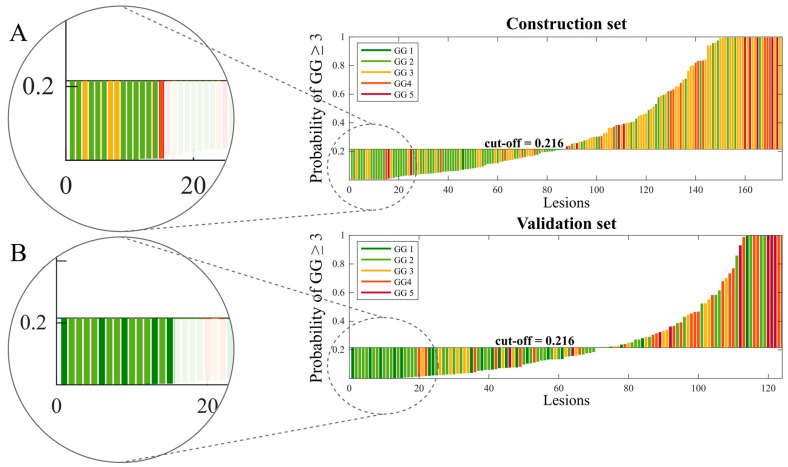
Waterfall plot of the output probabilities of the final classifier for construction set and validation set. The circles (**A**,**B**) highlight the lesions with zero likelihood of being high-aggressive. GG = grade group.

**Figure 5 cancers-16-00203-f005:**
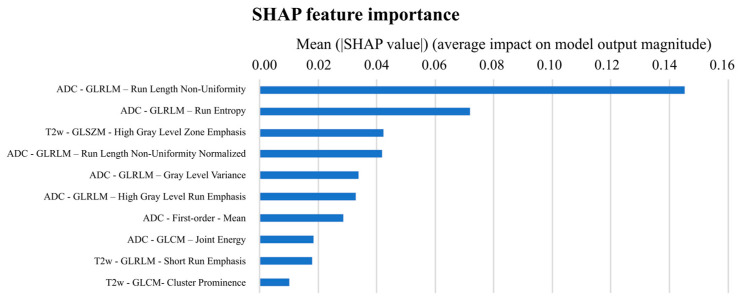
Ten features employed in the model ranked by their impact on model output. ADC = apparent diffusion coefficient, GLCM = grey-level co-occurrence matrix, GLRLM = grey-level run length matrix, GLSZM = grey-level size zone matrix, SHAP = Shapley additive explanation, T2w = T2-weighted.

**Figure 6 cancers-16-00203-f006:**
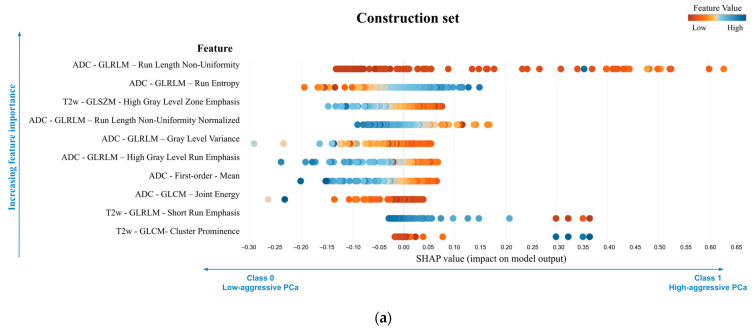
Shapley additive explanation (SHAP) summary plot for the construction (**a**) and validation (**b**) sets. ADC = apparent diffusion coefficient, GLCM = grey-level co-occurrence matrix, GLRLM = grey-level run length matrix, GLSZM = grey-level size zone matrix, SHAP = Shapley additive explanation, T2w = T2-weighted.

**Figure 7 cancers-16-00203-f007:**
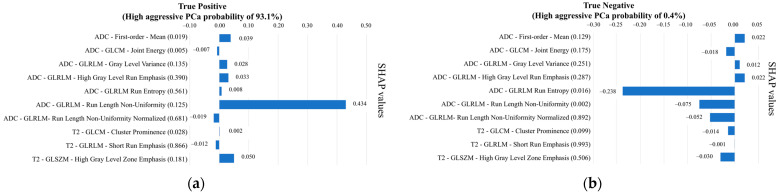
Shapley values of a true positive prediction (**a**) and a true negative prediction (**b**), randomly selected from the validation set. For each feature, the corresponding value is reported in parentheses. ADC = apparent diffusion coefficient, GG = grade group, GLCM = grey-level co-occurrence matrix, GLRLM = grey-level run length matrix, GLSZM = grey-level size zone matrix, SHAP = Shapley additive explanation, T2w = T2-weighted.

**Table 1 cancers-16-00203-t001:** Performances of the best model and results of the N−1 chi-squared test for the comparison of two proportions and of the DeLong test for the comparison of AUCs. Numbers in brackets represent the number of correctly classified cases over the total number of each class. Specifically, the resulting *p*-value, performances, and their differences are reported for centers A + B (construction set) and centers C + D (validation set) and individually for center C and center D. In bold, *p*-value < 0.05.

	Construction Set (Sample 1) vs. Validation Set (Sample 2)						Center C (Sample 1) vs. Center D (Sample 2)					
	AUC (95%CI)	Balanced Accuracy (%)	Sensitivity (%) (95%CI)	Specificity (%) (95%CI)	PPV (%) (95%CI)	NPV (%) (95%CI)	AUC (95%CI)	Balanced Accuracy (%)	Sensitivity (%) (95%CI)	Specificity (%) (95%CI)	PPV (%) (95%CI)	NPV (%) (95%CI)
Sample 1	0.75 (0.68–0.81)	72.2	70.4 [69/98] (60.3–79.2)	74.0 [57/77] (62.8–83.4)	77.5 [69/89] (69.8–83.7)	66.3 [57/86] (58.5–73.3)	0.77 (0.67–0.86)	69.4	55.2 [16/29] (35.7–73.5)	83.7 [41/49] (70.3–92.7)	66.7 [16/24] (49.5–80.3)	75.9 [41/54] (67.4–82.8)
Sample 2	0.73 (0.65–0.81)	67.9	61.8 [34/55] (47.7–74.6)	73.9 [51/69] (61.9–83.7)	65.4 [34/52] (54.7–74.7)	70.8 [51/72] (62.8–77.7)	0.63 (0.56–0.77)	59.6	69.2 [18/26] (48.2–85.7)	50.0 [10/20] (27.2–72.8)	64.3 [18/28] (52.0–74.9)	55.6 [10/18] (37.7–72.1)
*p*-value	0.731	0.423	0.278	0.989	0.120	0.546	0.143	0.274	0.290	**0.004**	0.857	0.103
|Diff| (95% CI)	0.02 (−0.1–0.1)	4.3 (−6.0–14.9)	8.6 (−6.5–24.1)	0.1 (−13.9–14.3)	12.1 (−2.9–27.6)	4.5 (−10.0–18.5)	0.15 (−0.1–0.3)	9.8 (−7.3–26.9)	14.0 (−11.3–36.7)	33.7 (9.95–55.2)	2.4 (−22.6–26.4)	20.3 (−3.4–44.1)

AUC = area under the receiver operating characteristic curve, CI = confidence interval, Diff = difference, PPV = positive predictive value, NPV = negative predictive value.

## Data Availability

The data presented in this study are available on request from the corresponding author. The data are not publicly available due to ethical reasons.

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
