# Peer review of "Development and Validation of an Explainable Radiomics Model to Predict High-Aggressive Prostate Cancer: A Multicenter Radiomics Study Based on Biparametric MRI"

_cancers, 2024, doi:10.3390/cancers16010203_

Round 1

Reviewer 1 Report

Comments and Suggestions for Authors

I have to admit that this is a manuscript of excellent quality and that it has very strong application potential. I will answer the questions one by one according to your requirements:

1. What is the main question addressed by the research?

A: This article is based on non-invasive diagnosis of prostate cancer grading in patients. Generally speaking, grading must be determined through sample pathology after RP or TURP. Typically, this does not affect the quality of care the patient receives. However, for advanced prostate cancer, it is very necessary to use endocrine therapy before RP or TURP. If the patient's grade or stage can be predicted through radiomics, this will be a very cost-effective diagnostic and treatment method for patients.

2. Do you consider the topic original or relevant in the field? Does it address a specific gap in the field?

A: Through literature search, there are many articles about prostate cancer radiomics. But this does not mean that this article does not have value. As I said, advance diagnostic prediction is very necessary for grading.

3. What does it add to the subject area compared with other published material?

A: They introduced the concept of Grade-Group for grouping. Although this is not the first time, its clinical value is very good.

4. What specific improvements should the authors consider regarding the methodology? What further controls should be considered?

A: This is an article with a relatively mature methodology, and the author has conducted multiple verifications, which is very important.

5. Are the conclusions consistent with the evidence and arguments presented and do they address the main question posed?

A: I think the results of multiple validation are very rigorous and accurate.

6. Are the references appropriate?

A: No self-references were found.

7. Please include any additional comments on the tables and figures.

A: Figure information is clearly expressed, and figure legend is clearly described. very good

For prostate cancer, imaging diagnosis is essential. Even with positive biopsy results, MRI results will affect subsequent diagnosis and treatment plans. This article recruited patients from 4 centers and used radiomics for cross-validation, which can ultimately accurately assess invasiveness in a non-invasive manner.

This article is innovative, fluent in writing, and rigorous in experimental design. The 5-fold cross-validation and validation set increase the reliability of the model.

It is recommended to accept.

Author Response

We thank the reviewer for her/his positive feedback. We are honored that she/he believes that our work has excellent potential. 

thank you very much for suggesting "acceptance"

Reviewer 2 Report

Comments and Suggestions for Authors

authors need to be congradulated on their effort for detailed documentation of imaging pre-pocessing and feature extraction procedures. The separation of model building and model testing is also very well thought. However, their are issues with both the design and methods.

1. for predictive modeling, base patient inclusion/exclusion criteria on outcome is generally prohibited. here authors only include everyone with biopsy proven PCa, which is unknown at time of MRI. this will introduce selection bias, the magnitude of such bias is unknown.

2. normalizing the T2W signal was understandable due to arbitrary magnitude, normalizing the ADC was not justified as absolute value of ADC has physical meaning. 

3. the methods mentioned in the feature selection section are all actually feature ranking not selection, AP clustering is unsupervised and the rest are supervised. there are many other methods commonly used in the literature (e.g. lasso logistic, random forest...), the justification of using these methods are unclear. especially lasso logistic regression is a proven method that provides simultaneous variable selection and model fitting. It usually performs well and has be widely used. on the other hand, methods like naive bayesian requires unique assumption on independence and prevalence.

4. terminology wasn't clear, what is generalized logistic regression? logistic regression is one specific form of generalized linear regression.

5. by doing 5-fold cv inside the construction set and test in the external validation test, author effectively performed 1 step of the outer loop in a nested cv design. there is lost opportunity for a true nested cv that construct the model based on permutations of two centers and test on the other two centers.

6. Youden's index is just the sum of sensitivity and specificity. there is no justification that maximizing the sum of sensitivity and specificity is a optimal strategy in the clinical setting. Formal decision analysis is necessary if any classification cut-off value is proposed.Furthermore, due to finite data, all sensitivity and specificity pairs are discrete, the cutoff point associated with the max Youden's index will depend on what method was used to estimate the ROC curve and the smoothing technique (mainly non-parametric step-function vs binormal, but other less common methods also exist).

7. why only balanced accuracy was reported but not balance PPV and NPV? just reporting AUC is good enough. author need to also plot the ROC curves, some model may be more sensitivity and some may be more specific in general.  

8. I dont think chi2 test is approperiate for comparing two AUC in table 1.

Author Response

We thank the reviewer for carefully reading our paper and for acknowledging our effort in designing the experiments. We believe that thanks to her/his comments the paper has increased its potential. We will answer the comments point by point in the attached file

Reviewer 3 Report

Comments and Suggestions for Authors

In this manuscript, the authors propose a radiomics model based on biparametric MRI to distinguish between low- and high-aggressive prostate cancers. The results are comparable to existing methods, and they further validate the model on external dataset and provide an explainable approach. The model appears promising and has potential to benefit clinical practice. However, there are several points need to be addressed.

1. The authors claims that all 15 lesions in the validation set with a likelihood of being high aggressive equal to zero were indeed GG1/2. However, for the construction set the number is 11/15. It is hard to say the model is sufficient to identify low-aggressive PCa. Also, the author should explain what causes this difference in the construction set and validation set.

2. The author say that when assigning a high likelihood (>70%) on the external validation set, only one GG1 was misclassified. However, there are also several GG2 lesions in likelihood > 70%, as seen in Figure 3B. It is very confusing since the aim of the paper is to distinguish GG1/2 and GG>2. Also, if including these GG2 lesions, the effectiveness of this model need to be further investigated.

3. In line 266 in the Discussion, where does a PPV of 94% (16/17) come?

4. In Figure 5, why do some features have mainly low or high feature values, as seen by the color? And does the feature value distribution affect the feature importance in this model?

5. In the explanation of the model, the authors present the SHAP results. More discussion on the cause of the feature importance should be given and further explain the feature impact to PCa aggressiveness according to the literature.

Comments on the Quality of English Language

The English need to be checked carefully. There are a few typos and grammar mistakes. 

Author Response

We thank the reviewer for carefully reading our paper and acknowledging that it is promising work that might have the potential to benefit clinical practice. We will address reviewer’s concerns in the attached point-by- point response.

Round 2

Reviewer 2 Report

Comments and Suggestions for Authors

authors answered some of my comments but not all. Here are my clarifications.

1. based on the exclusion criteria, patients with MRI but no path results were excluded. there may be underlying reason why this happens and there is no documentation on how many are there. this may results in underrepresentation of GG0 or GG1 studies in either or both the construction and validation dataset. Any shitfs between institutions will affect model performance and generalizability. this is the bias i was talking about, please at least list it as one of the limitation.

2. author wrote "the parameters of the models were tuned, and the classification cut-off was optimized based on the Youden index" in the method section. however, models like logistic regression and lasso regression output probabilistic predictions and parameters were estimated based on maximize the likelihood function (with penalty in the case of lasso) or metrics such as deviation and area under the ROC curve. It is non-trivial to modify that to maximize youden's index. I think author obtained the cutoff by maximizing youden's index in the probabilistic predictions after estimating the model parameters instead of directly estimating the parameters by maximizing youden's index. 

3. author need to use AUC as main assessment of diagnostic performance because it does not rely on any cutoff. either remove youden's index (and any cut-off analysis) from manuscript or only report as one example cutoff. In general, the use of youden's index over other cutoff selection methods is arbitrary, it should not be labeled as an "optimized" cutoff, it is just one cutoff. A cutoff can be called "optimized" when it maximizes a clinically meaningful metric by weighting false positives and false negatives appropriately in a decision analysis framework (which may change as population changes). (balanced) accuracy, sensitivity and etc. are all affected by cutoff therefore not very informative. 

4. author's new discussion on selection cutoff based on NPV is not appropriate. maximizing just the NPV is equivalence to a "treat-none" decision. any cutoff should be a trading between fp and fn. so the appropriate strategy for finding a cutoff is to first estimate how many false negative(s) is clinically tolerable for one false positive in a setting that favor the detection of negatives. constructing the corresponding cost function and a cutoff can be derived by minimizing that cost function. I think this discussion is out of the scope of the current manuscript. 

5. i was referring to ROC curves for table A4. please add ROC curves for the top models. 

6. based on the low sensitivity and specificity, and naive bayesian (high bias low variance model) out performed the Lasso (low bias high variance model), the study sample seems to be noisy and with low information. predicting high-aggressive prostate cancer with radiomics remains very challenging.

Comments on the Quality of English Language

i would be satisfied if author report ROC and AUC for model performance and reduce or remove any cutoff related analysis. 
